# Deep Endometriosis and Infertility: What Is the Impact of Surgery?

**DOI:** 10.3390/jcm11226727

**Published:** 2022-11-14

**Authors:** Angelos Daniilidis, Stefano Angioni, Stefano Di Michele, Konstantinos Dinas, Fani Gkrozou, Maurizio Nicola D’Alterio

**Affiliations:** 1Department of Obstetrics and Gynecology, Hippokratio Hospital, Aristotle University of Thessaloniki, 546 42 Thessaloniki, Greece; 2Department of Surgical Science, University of Cagliari, Cittadella Universitaria Blocco I, Asse Didattico Medicna P2, Monserrato, 09042 Cagliari, Italy; 3University Clinic in Obstetrics and Gynecology, University of Ioannina, 451 10 Ioannina, Greece

**Keywords:** deep endometriosis, infertility, surgery, ART, MAR

## Abstract

In women with deep endometriosis, the spontaneous fertility rate might range from 2 to 10%. The optimal management of these women is still an area of debate. Therefore, this review aims to explore the literature on the impact of deep endometriosis surgery on reproductive outcomes and pregnancy rates in women with and without prior infertility. A total of 392 articles were identified through database searching. Twenty-three studies were eligible to be included in the review. A total of 1548 women were identified, 814 of whom became pregnant, with a mean pregnancy rate of 52.6% (95% CI 49.7–63%). Our review suggests that surgery may improve fertility outcomes. Due to the variability in the studies, it is impossible to stratify fertility outcomes of surgery by the localization of deep endometriosis. More investigations are needed to determine whether surgical management should be first-intention or limited to the failure of medically assisted reproduction treatment.

## 1. Introduction

The histological analysis of an endometrial-like sample (consisting of glands and endometrial stroma) expanding out of the uterine cavity, most frequently implanted in the peritoneal cavity, are the hallmark of endometriosis. This chronic, inflammatory, hormone-influenced disorder typically affects women in their reproductive years [1,2]. Recent studies have shown that endometriosis has a wide range of etiological factors, with its origins in genetics, metabolomics, microbiomics, immunology, endocrinology, and particular environmental implication [3,4,5,6,7,8,9]. Endometriosis can clinically appear as microscopic, subtle, typical, cystic ovarian, or deep endometriosis lesions, according to a recent review by Koninckx et al. [10]. Affected patients have a wide range of severe symptoms, most of which are concentrated in the pelvis and lower abdomen (such as chronic pelvic pain, dysmenorrhea, dyspareunia, dyschezia, and dysuria) [11], which have a major impact on women’s wellbeing and quality of life [12,13,14,15]. Its overall prevalence is around 7–10%; however, in women who experience pain or infertility, deep endometriosis in the severe pattern is more prevalent than 20% of cases [10,16,17,18]. The main location of deep endometriosis include the torus uteri, posterior fornix, uterosacral ligaments, vagina, bowel, and urinary tract [17]. Deep endometriosis can be treated medically or surgically, depending on the clinical situation and the patient’s desires [3]. Medical treatment (e.g., progestins, gonadotropin releasing hormone agonist and antagonist) is beneficial in stopping the development of lesions and inducing their regression, leading to improved symptoms [19,20,21,22]. Patients who do not respond to medical treatment and have severe symptoms (such as hydronephrosis or intestinal obstruction) may require surgical intervention [23,24]. However, surgery can be stressful, since the highest level of expertise is needed and can include many risks [25]. The intervention may cause injury to the bladder, ureter, and intestines, which could result in challenging-to-treat short-term scenarios such as hemorrhage and ureteral injury, as well as long-term complications such as ureteral or rectal stenosis, fistulas, or neurologic deficits of the bladder or rectal function. These problems occur between 1 and 10% of the time [25]. Furthermore, surgery does not offer protection against recurrences. 

Up to 50% of women who experience infertility have endometriosis. Endometriosis is 10 times more common in infertile women than in the general population, with a prevalence of 25–40% [26,27,28,29,30,31,32,33,34]. In women with severe endometriosis, the spontaneous fertility rate might range from 2 to 10% [35]. Specifically, in women with deep endometriosis, gamete migration and tubal function can undoubtedly be impaired by the irregular anatomy, extensive intra-abdominal adhesions, and the chronic inflammatory environment [36,37]. Given the well-established connection between endometriosis and adenomyosis and the potential connection between both conditions and infertility, the scenario becomes even more problematic [38,39]. The literature shows that adenomyosis is found more frequently in women with deep endometriosis than those with other disease presentations. In fact, Vercellini et al. demonstrated that concomitant adenomyosis in patients with deep endometriosis significantly reduces the likelihood of conception following extensive endometriosis surgery [40]. 

If deep endometriosis triggers infertility, it makes sense to assert that surgically removing it should increase fertility. This has been a cornerstone of endometriosis therapy for many years, but it is currently under growing investigation. According to Adamson et al.’s meta-analysis, surgical excision of lesions in women with ASRM stage III-IV was linked to a higher spontaneous pregnancy rate, which improved from 4% preoperatively to 43% postoperatively with a differential gain in pregnancy of 39% [41,42]. Measuring the effects of surgery on fertility in women with deep endometriosis is challenging because many women undergo surgery because of pain, with fertility being a secondary issue. Although there is no convincing evidence that operative laparoscopy for deep endometriosis improves fertility (because there is no RCT comparing reproductive outcomes after surgery and after expectant management), the ESHRE 2022 guidelines state that operative laparoscopy may be a treatment choice for symptomatic patients who want to become pregnant [41]. If there is bowel or urinary obstruction or if the pain is persistent and severe and not responding to medical treatment, deep endometriosis surgery is necessary and recommended [41]. Conversely, surgery is linked to lifelong problems and de novo adhesions that could harm the success of natural pregnancy or obtained via medically assisted reproduction [43]. When having to decide whether to perform surgery, the guidelines development group (GDG) advises that factors such as the presence or absence of symptoms, history of prior surgery, patient age and preferences, ovarian reserve, the presence of other infertility factors, the duration of infertility, and the estimated endometriosis fertility (EFI) index should all be taken into account [41]. The best way to treat these women is still debatable, particularly when asymptomatic patients’ fertility is the main priority. Additionally, as in previous guidelines, there is no evidence to support the benefits of surgical excision of deep endometriosis prior to medically assisted reproduction; hence no recommendations could be provided. However, the GDG stated in the 2022 ESHRE guidelines that patients who would benefit from medically assisted reproduction after surgery should be identified using the EFI score and that women should be advised about the risk of falling pregnant after surgery. Other reproductive tests, such as their partner’s sperm analysis, should be explored [41]. 

It is necessary to understand the effect of surgery for deep endometriosis on fertility as long as current evidence suggests a role for surgery in patients with deep endometriosis who want to become pregnant [44]. To achieve this, it is important to distinguish reproductive outcomes according to deep endometriosis localization, the type of surgery, the presence of proven preoperative infertility, and the intervention of medically assisted reproduction. Therefore, this review aims to explore the literature on the impact of deep endometriosis surgery on reproductive outcomes and pregnancy rates in women with and without prior proven infertility. Additionally, we reviewed data according to specific surgical procedures, disease localization, or patient subgroups (patients with specified previous infertility before the surgery or women who desire to conceive before and after the surgery). Finally, we collected information on safety challenges and details about surgical complications.

## 2. Materials and Methods

A systematic electronic database search of all published studies, limited to the English language, on the effect of surgery for deep endometriosis on fertility from January 2010 to June 2022 was performed in PubMed, MEDLINE, Embase, Web of Science and the Cochrane Library. The Preferred Reporting Items for Systematic Reviews and Meta-Analyses (PRISMA) guidelines were followed [45]. We used combinations of the following Mesh keywords in the search: “deep endometriosis”, “deep infiltrating endometriosis”, “deep infiltrative endometriosis”, “intestinal endometriosis”, “bowel endometriosis”, “colorectal endometriosis”, “rectovaginal endometriosis” “uterosacral endometriosis”, “vaginal endometriosis”, “bladder endometriosis” were combined with “surgery” AND “fertility”, “infertility”, “in vitro fertilization (IVF)”, “Assisted Reproductive Techniques (ART)”, and “medically assisted reproduction”. 

Only publications containing specific information on pregnancy rates in patients operated on for deep endometriosis were included. The postoperative spontaneous pregnancy rate or medically assisted reproduction rate and total pregnancy rate were reported. Total pregnancy rate was defined as the number of women achieving pregnancy with or without medically assisted reproduction treatment. Moreover, articles were categorized based on the definition of the total number of women seeking pregnancy after surgery, the number of infertile patients before surgery and the modality for achieving conception (spontaneous or with medically assisted reproduction attempts). We also extracted data on infertility (defined by the absence of conception after one year of attempt and/or premature failure of ART) specifically including in a subgroup only patients with proven infertility before the surgery who attempt to conceive after the surgical treatment. The year of publication, setting, number and clinical characteristics of recruited subjects, surgical approach, localization of deep endometriosis lesions, type of surgical treatment, and live birth rate (LBR) were reported. According to the treated compartment, specific data on pregnancy outcomes concerning the operative complications and recurrences of the disease were explicitly searched for. 

After an initial screening of titles and abstracts retrieved by the search, the full texts of all potentially eligible studies were fully assessed. Only papers in English were included. To ensure the relevance of the publications retrieved, additional inclusion criteria were applied: to be included, the published studies had to contain a clear description of fertility outcomes after surgery. Redundant articles were removed after an initial selection, and then other articles were removed if their title, abstract, or material and methods did not fit the objective of our review. The references of the included articles were also reviewed, and additional studies were added if relevant. If there were several publications on the same patient series developing over the years, only the latest was included. Studies were also excluded if data from the same or similar series were reported repeatedly. The full texts were examined for eligibility, and articles satisfying the abovementioned criteria were selected. We included prospective and retrospective cohort, case–control, and observational studies. Only two prospective cohorts in randomized control trials (RCT) were included. Reviews, case reports, and abstracts of scientific meetings were not included. The primary outcome was the total pregnancy rate after the surgical procedure. The secondary outcomes were: pregnancy rate in the subgroups of infertile patients seeking a pregnancy, pregnancy rate according to the main localization of the disease, and surgical technique performed and complications occurred. 

A reviewer (M.N.D.) followed specific parameters to collect the data, and a second reviewer (S.D.) individually repeated the process for the whole set of included publications. No differences were discovered when the data retrieved by the first and second reviewers were compared. A third reviewer (S.A.) would have been consulted if any differences had been discovered. We investigated all patients who had undergone surgery for deep endometriosis regarding fertility and pregnancy outcomes. To study various scenarios, we also performed data analysis according to specific surgical procedures or patient subgroups (deep endometriosis with or without bowel or bladder involvement). 

The terms of medically assisted reproduction and assisted reproductive technology (ART) used in this review are those provided by the World Health Organization’s International Committee Monitoring Assisted Reproductive Technologies (WHO ICMART) [46]. Ovulation induction, controlled ovarian stimulation, ovulation triggering, ART procedures, and intrauterine, intracervical, and intravaginal insemination with the husband/or partner’s donor’s semen are all considered part of medically assisted reproduction. Intrauterine insemination (IUI) and ART are thus included in medically assisted reproduction. All treatments or procedures that include managing human oocytes, sperm, or embryos in vitro to conceive are referred to as ART. 

The majority of deep endometriosis surgery studies include fertility as a secondary objective. As a result, information about the methods utilized to achieve pregnancy was frequently unavailable. No statistical comparisons were done on outcome variables due to the study groups and design heterogeneity. All provided data were evaluated as descriptive, and results are given as an overall mean with its computed confidence interval for all the women studied in this review.

## 3. Results

The selection and elimination of the articles are detailed in the flow chart in Figure 1.

Despite finding 392 publications, our selection criteria limited the number of papers for analysis to 23. After removing duplicates and screening only prospective and retrospective cohort, case–control, and observational studies, 88 out of 134 were excluded as they were deemed irrelevant by observers (e.g., if only technical details or effect on pain were reported). In the end, only 23 studies clearly described fertility outcomes after surgery. These publications are divided into 15 retrospective observational studies, 2 prospective observational studies, 3 retrospective controlled studies, 1 prospective controlled study, and 2 randomized controlled trials (RCT). Except for the two RCTs, all the studies have a high risk of selection and allocation bias. Only one retrospective study [47] statistically controlled the latter risk using propensity scores to compensate for potential allocation bias. We included only studies where a complete or incomplete surgery was performed before the assessment of the pregnancy rate. Rubod et al. [48] compared three groups of patients: complete surgery of deep endometriosis lesions, incomplete surgery, and no surgery before ART. Only complete and incomplete surgery group data were considered. Bendifallah et al. [47] compared the impact of first-line ART and first-line surgery for colorectal deep endometriosis followed by ART on fertility outcomes in women with deep endometriosis-associated infertility. Only women who underwent surgery before ART were included in our analysis. In most studies, the number of patients enrolled differed from those followed during follow-up to investigate fertility. Only the patients whose outcomes of pregnancy were analyzed were considered. 

Details of the characteristics of the selected studies and fertility outcomes of the women who underwent surgery for deep endometriosis are shown in Table 1. 

A total of 1548 women were identified to investigate fertility outcomes, 845 of whom have a documented infertility before surgery and 1081 of whom wished to become pregnant without a diagnosis of infertility. The number of subjects to evaluate pregnancy rate varied from 9 [54,63] to 180 [64]. In the entire group of 1548, 814 women became pregnant, with a total pregnancy rate varying from 18.7% [52] to 83.8% [69], with a mean of 52.6% (95% CI 49.7–63%). When reported, in the studies included in our review, we found 946 women who sought to conceive (followed to investigate fertility), of whom 501 became pregnant with a mean of (53%). From a clinical point of view, it is necessary to distinguish patients with true infertility (no pregnancy after one year of trying) from those wishing to conceive without proven infertility. Among the 23 studies evaluating fertility after surgery for deep endometriosis, it was possible to distinguish between true infertility and those wishing to conceive without proven infertility in some series. A sub-analysis was performed to investigate pregnancy rate in in those women who had a diagnosis of infertility before surgery and who specifically sought a pregnancy after the surgery (Table 2): in this group of 635 women, 337 patients achieved pregnancy, with a total pregnancy rate of 53%, 95% CI (46.3–63.4), of whom 144 (42.7%) were spontaneous and 193 (57.3%) were obtained via medically assisted reproduction. In this table, the rates of spontaneous pregnancy and medically assisted reproduction are conditioned by the fact that the three studies included [47,48,62] are focused on the outcomes only in patients treated with ART after surgery. If we removed these three studies from the table, the total pregnancy rate, spontaneous pregnancy rate, and medically assisted reproduction rate would be 53.2%, 38.5%, and 14.7%, respectively.

Moreover, the results on pregnancy rate, postoperative complications, and recurrences have been separated according to the compartment involved in deep endometriosis. Table 3 shows the results after colorectal surgery for bowel deep endometriosis. Total pregnancy rate after surgery ranged from 25 [57] to 84% [69], with a mean of 50.3% (95% CI 47.2–65.6%) for the twelve studies added together. Spontaneous pregnancy rate ranged from 0% in the two studies focusing only on ART results [47,62] to 61.1% [58]. The pregnancy rate obtained via medically assisted reproduction ranged from 8.3 [58] to 60% [62]. The complication rate described in different studies occurred concerning Grade III, according to Clavien–Dindo classification [70], in 7.45% of women, including 2% of women experiencing anastomotic leakage (ranging from 1.3 [53] to 3.6% [55]), 2.4% experiencing rectovaginal fistula (ranging from 0.7 [66] to 19% [63]), and 5.2% experiencing bladder atony requiring self-catheterization for at least 12 weeks (ranging from 0.6 [57] to 12.7% [69]). Recurrences, when described, occurred in 4.8%, depending on the follow-up period, ranging from 3.6 (follow-up of seven years) [69] to 6.7% (follow- up of about five years) [57]. 

In Table 4, a sub-analysis of the studies on colorectal deep endometriosis was performed, separating pregnancy rate in patients who underwent conservative surgery (shaving or discoid resection (DR)) versus patients who underwent segmental resection (SR) showing a total pregnancy rate of 69% and 45%, respectively.

As shown in Table 5, focusing on outcomes after surgery for bladder deep endometriosis, the five studies indicated a total pregnancy rate of 65.9% (95% CI 54.5—74.2%), of which 55.5% were spontaneous and 44.5% were obtained with ART. Grade III complications occurred in 7.2 %, and recurrences were described only in one study with a rate of 2.9% [59].

Finally, Table 6 shows the pregnancy rate, complications, and recurrences in articles focusing on deep endometriosis without specific involvement, showing a total pregnancy rate of 52.7, a spontaneous pregnancy rate of 24%, and a pregnancy rate achieved with ART of 29%. Grade III complications and rectovaginal fistula occurred respectively in 7.9 and 1%.

## 4. Discussion

In addition to the characteristics and diffusion of the endometriosis lesions, the surgical management of different forms of endometriosis largely depends on the woman’s general characteristics (age, parity, future wish for conception, duration of infertility, and symptoms) [71]. These variables influence the type of surgical intervention and the development of personalized therapy strategies from adolescence to menopause. These strategies aim to increase the quality of life and chances of future pregnancy while lowering the risk of recurrence over time [72]. The surgeon must be careful about the uterus, fallopian tubes, and ovaries to preserve a young woman with endometriosis’s chances of conceiving successfully after surgery. Following surgery, conception can occur naturally or via medically assisted reproduction [73]. Due to the variability in the studies, it is impossible to evaluate the validity of surgery to improve fertility according to deep endometriosis locations. Identifying a homogeneous patient population with a single deep endometriosis location is very difficult. The fertility outcome is often a secondary objective in the majority of the studies with retrospective nature and non-controlled design. The primary objective is to determine the impact of surgery on symptoms, postoperative complications, and changes in quality of life. Generally, limited information was available on the duration of infertility and the coexistence of additional infertility factors other than endometriosis. Moreover, most articles did not report how many patients underwent surgery because of pain, infertility, or both.

Comparing our results with a mean of total pregnancy rate of approximately 53% following the intervention with those of other systematic reviews on this topic, it can be seen that in Vercellini et al., the overall weighted mean of pregnancy rate after excisional surgery for deep endometriosis (in particular rectovaginal endometriosis) was slightly over 40% [43]. Another systematic review, dividing the studies into those with and without bowel involvement, found overall pregnancy rates (OPR) after surgery of 46.9 and 68.5%, respectively. The spontaneous pregnancy rate after surgery was 50.5% in patients without bowel involvement and 28.6% in those with colorectal deep endometriosis [74]. In their review, Iversen et al. included four retrospective and three prospective observational, uncontrolled studies on colorectal surgery for deep endometriosis. The postoperative spontaneous pregnancy rate and OPR were 49 and 63% in retrospective studies and 21 and 55% in prospective studies, respectively [75]. In another review on fertility outcomes (spontaneous and via medically assisted reproduction) after colorectal surgery, including 26 series published between 1990 and 2015 involving a total of 855 women, the overall pregnancy rate was 51.1% (spontaneous pregnancies 31.4% and medically assisted reproduction 19.8%) [76]. In this context, it must be underlined that these data may overestimate the advantages of surgery and that it is difficult to attribute this success rate only to surgery. Most case studies involve pregnancies achieved with ART and women who were not infertile before the intervention. Exclusively natural pregnancies should be considered when determining the precise fertility-enhancing effect of deep endometriosis removal. Furthermore, in order to provide a valid comparison with first-line ART, the pregnancy rate following surgery should only include infertile women [77]. Furthermore, it might be challenging to evaluate fertility outcomes following deep endometriosis surgery because most authors fail to distinguish between truly infertile women (any attempt to become pregnant more than a year before surgery) and those seeking to become pregnant but are not infertile. Interestingly, Vercellini et al. repeated a systematic review on this issue. Still, they only included studies that reported the probability of pregnancy in infertile women with deep endometriosis, only being focused on natural conception. Developing results were less encouraging, with a standard estimate of the success rate of 24% (IC95% 20–28%) and a differential gain in pregnancy of 15%, reducing the spontaneous pregnancy rate from an initial 39% when all patients were taken into account regardless of preoperative fertility status or the application of ART [78]. Data consistent with our findings are presented in the sub-analysis of infertile patients (Table 2), with a fall in pregnancy rate from 52.6% to 38.5%, and in the review of Darai et al., with a decrease from 51.1% to 31.4% [76].

### 4.1. Fertility in Women Who Underwent Colorectal Surgery for Deep Endometriosis

In women with intestinal deep endometriosis, fertility must be carefully examined, especially following surgery. Confounding aspects of infertility, such as the surgical approach, also had to be considered. Surgery for bowel endometriosis involves different procedures, including DR, SR, and rectal shaving [57,58]. Previous studies highlighted the importance of the surgical approach for bowel resection in fertility outcomes, showing that women who underwent open surgery had a greater prevalence of grade III postoperative complications but no major differences in spontaneous pregnancies occurred following open o laparoscopic surgery [57,58].

#### 4.1.1. Fertility in Women Who Developed Severe Complications after Colorectal Surgery

Regarding complications, Ferrier et al. described reproductive outcomes of a group of patients who underwent colorectal surgery for endometriosis developing severe complications, including rectovaginal fistula, anastomotic leakage, deep pelvic abscess, ureterohydronephrosis, urinary fistula, and bowel obstruction (grades III–IV of the Clavien–Dindo classification). A total of 48 women considered natural conception after surgery, with 16 (33%) conceiving after a median of three years. After complications of grades IIIa and IIIb, the pregnancy rate was 66.7% and 40%, respectively. Any woman who suffered from a grade IV complication became pregnant [79]. It would emerge to be essential to recognize women who have a low chance of spontaneous pregnancy after a grade III–IV complication in order to offer medically assisted reproduction treatment as soon as possible because a lower cumulative pregnancy rate (CPR) was discovered for those who experienced anastomotic leakage (with or without rectovaginal fistula) or deep pelvic abscess [79]. In our series, complication rates occurred with a mean of approximately 7–8%. In the systematic review of Iversen et al., major postoperative complications were observed in 9% of patients included in the retrospective studies and 13% of those included in prospective studies. Anastomotic leakage that needs reintervention was reported in 5% of the women who underwent segmental colorectal resection [75].

#### 4.1.2. Choosing the Best Treatment for Removing Bowel Deep Endometriosis to Improve Pregnancy Rate

A series of studies included in this review explored the best treatment for removing colorectal deep endometriosis (conservative surgery or segmental resection) to improve pregnancy rate. While Meuleman [56] and Hudelist [66] found no differences in pregnancy rate (both spontaneous and ART) between performing segmental resection or not, and choosing a conservative treatment, Ballester et al. reported ART outcomes in 60 infertile patients after surgery for colorectal deep endometriosis, demonstrating a decrease in CPR for women after SR compared to rectal shaving or DR. Interestingly, a trend toward a lower CPR was seen in women who started their first IVF cycle more than 18 months after surgery [62]. However, it is challenging to compare the two groups because women who underwent SR undoubtedly had larger nodules removed. Hudelist et al. and Meuleman et al. both had similar clinical outcomes, except for a greater minor complication rate in patients undergoing SR [56,66]. The results of our analysis are consistent with those of Ballester et al. because the group of patients who underwent SR had a lower total pregnancy rate and spontaneous pregnancy rate of 45 and 36.7%, respectively, than conservative surgery patients with a greater total pregnancy rate and spontaneous pregnancy rate of 69.2 and 46.2%, respectively. 

If we only consider the ART outcomes (as in the study of Ballester et al.), pregnancy rates do not differ between the two groups (23 versus 21.4%), even in the results of our review.

Finally, Roman recently published the results of a seven-year follow-up of patients participating in the ENDORE RCT, in which patients were randomly assigned to either SR or nodule excision via shaving or DR [69]. The long-term follow-up verified what had already been reported in their 5-year follow-up [80], i.e., that functional results, pregnancy rate, and deep endometriosis recurrence risk were similar in women treated with either the conservative or radical technique. A total pregnancy rate of 83.8% was observed, separated into 82.4% and 85% in the two arms, and 47 pregnancies in 31 women were reported, with 57.5%resulting from natural conception and 75.7% overall live births rate (OLBR) [69]. These findings were among the highest previously documented in the literature, with more than half of the conceptions being natural, demonstrating the benefit of colorectal deep endometriosis surgery on fertility as well as symptom relief. These findings rely on the length of follow-up (a longer follow-up gives women more time to conceive, increasing the pregnancy rate), other concomitant endometriosis locations (particularly adnexal involvement), the presence of uterine adenomyosis (found in most women who undergo surgery for rectal endometriosis), and a preoperative history of infertility [73]. 

### 4.2. Fertility in Women Who Underwent Bladder Surgery for Deep Endometriosis

Considering bladder endometriosis, another study that reported a similar high pregnancy rate to Roman et al. was a retrospective cohort analysis that discovered an 81% of total pregnancy rate after partial cystectomy or partial-thickness detrusor muscle excision [59]. Similarly, in our series of women with bladder endometriosis, posterior compartment involvement and associated treatment are not fully reported, and there is no analysis of reproductive outcomes based on surgical management of bladder endometriosis (bladder resection vs bladder shaving with and without detrusor closure). Another key challenge with bladder endometriosis and infertility is determining whether posterior deep endometriosis should be removed simultaneously. Timoh et al. found that the pregnancy rate was high and comparable in both the group that underwent just a partial bladder resection and the other group that underwent additional procedures on the posterior compartment, indicating that this approach had no harmful effects on fertility [65]. Saavalainen et al. found similar pregnancy rate of 67% and 62% when comparing patients with bladder endometriosis alone and those with concomitant surgeries. These facts demonstrate that simultaneous removal of the anterior and posterior deep endometriosis has no harmful effects on fertility. However, this approach offers a higher risk of complications, particularly voiding dysfunction [60]. 

### 4.3. Pro and Cons of First-Line Surgery

First-line surgery as a primary treatment could be favored in patients with severe pain symptoms, without tubal factors, and with a normozoospermic partner [77]. According to the ESHRE guideline 2022 on the management of women with deep endometriosis, stated that “Although no compelling evidence exists that operative laparoscopy for deep endometriosis improves fertility, operative laparoscopy may represent a treatment option in symptomatic patients wishing to conceive” (ESHRE 2022 guidelines, p. 10) [41]. Symptoms could mean not only acute or chronic pain but also deep dyspareunia, silent kidney (never painful), neuropathy such as sciatic pain, dysuria or bloating with abdominal discomfort, which carries a risk of bowel occlusion [44]. The ENDORE RCT showed that pregnancies are attainable following surgery for deep endometriosis, with a total pregnancy rate of more than 80% and more than 50% of spontaneous pregnancy rate. Additionally, in the subset of infertile patients who had preoperative conception failure for more than 12 months, 74% achieved pregnancy with 53% of natural conceptions, allowing us to conclude that surgery may consider ART unnecessary in one out of two women in addition to treating the symptoms [69]. Additionally, there is no increased chance of multiple gestations following spontaneous conceptions. As a result, surgery may be the most effective treatment for pain and fertility when infertility is combined with pelvic pain. 

Avoiding the overestimation of the surgical effect is crucial from a therapeutic standpoint since, in contrast to treatments for other endometriosis forms, radical removal of intestinal endometriotic nodules can negatively affect women’s health, as well as future fertility [81]. In the case of deep endometriosis, surgery is related to serious lifetime complications such as rectovaginal or urinary fistulas, abdominal abscess, intestinal obstruction, or stenosis, all of which can negatively influence patients’ quality of life. In addition, while surgery may excide all the endometriotic lesions, it often cannot restore the anatomy and physiology to achieve the complete removal of the lesions because some may be hidden or technical difficulties may occur, exposing the patients to a risk that may not be justified in terms of a risk/benefit ratio. Adhesion recurrence, in particular, is quite common following a surgical operation [82]. 

### 4.4. Surgery before or after ART 

There is no evidence in the literature to suggest the surgical removal of deep endometriosis prior to ART in infertile women with endometriosis to improve reproductive outcomes. Because no RCTs are yet published, no definitive conclusion can be provided. ESHRE guidelines only propose using the EFI score, even it should not be used as a predictive tool. Women with a high EFI score (7–10) may be encouraged to choose spontaneous pregnancy; those with an intermediate score (5–6) may be advised to choose either spontaneous pregnancy or medically assisted reproduction depending on age and ovarian reserve; and those with a low score (5) should receive medically assisted reproduction treatment [83]. The theory that states there should be no surgery before ART has some exceptions, such as hydrosalpinges shown on ultrasounds, which are known to change the outcome of ART with an approximately halved chance of conception. If hydrosalpinges are present, they must be removed or clipped in cases where salpingectomy is technically difficult or impossible [84]. 

In a meta-analysis, Casals et al. [85] evaluated the reproductive outcomes between patients who received IVF without having first undergone deep endometriosis surgery and those who did. Compared to women who received IVF without surgery, the results demonstrated a statistically significant advantage for surgery for deep endometriosis prior to IVF, with a pregnancy rate per patient and pregnancy rate per cycle 1.84 times more likely. In particular, in one of the studies included both in our review and in this meta-analysis, Bendifallah et al. [47] identified a group of women with a favorable prognosis to become pregnant following IVF (age < 35 years, anti-Mullerian hormone (AMH) > 2 ng/mL, and no adenomyosis). In a subset of women with at least one unfavorable characteristic, women who underwent first-line surgery had a considerably higher pregnancy rate. In the sample of women with favorable prognostic characteristics and AMH serum concentration less than 2 ng/mL, cumulative LBRs were significantly higher for women who underwent first-line surgery followed by ART than those who received first-line ART. In this scenario, Ballester et al. [86] discovered that an AMH level < 2 ng/mL was significantly correlated with a lower chance of achieving pregnancy after IVF-ICSI. Stochino-Loi et al. [64] conducted a retrospective study to see if surgery for severe endometriosis could be recommended in women with low (1–1.99 ng/mL) and very low AMH level (1 ng/mL) ovarian reserve with good pregnancy outcomes when compared to postoperative pregnancy rate in women with normal AMH level (>2 ng/mL). There was no statistically significant difference in the pregnancy rates of women with normal and low AMH levels, 74.6% and 73.9%, respectively. This study showed that preoperative AMH levels did not significantly affect the likelihood of the postoperative pregnancy rate between the groups that successfully became pregnant. Low AMH levels have been linked to lower pregnancy rate in ART, according to numerous studies [87]. More precisely, low AMH concentrations were linked to a reduced ability of the ovaries to respond to hyperstimulation, which caused decreased rates of pregnancies after ART [88]. In contrast, the link between AMH level and spontaneous conception rate appears to be debatable, as women with low AMH achieved a satisfactory spontaneous pregnancy rate [89]. These last findings convinced us that recovering the ability to conceive naturally through surgery would be an alternative solution to primary ART in women with low AMH. We believe surgical intervention should not be avoided in women with low AMH levels since spontaneous conception may counterbalance the poor response to ovarian hyperstimulation for IVF. However, when surgery and IVF are combined, it appears challenging to separate the influence of the two procedures without undertaking an RCT comparing surgery plus postoperative IVF with IVF alone. With the results of the ENDOFERT trials (https://www.clinicaltrials.gov/ct2/show/NCT02948972, accessed on 5 July 2022), this kind of information should be available in the following years. 

Since Ballester et al. [86] showed that only a small number of pregnancies were acquired over two IVF cycles in women with colorectal deep endometriosis, surgery may also play a role following IVF treatment. Therefore, there may be a place to evaluate whether removing deep endometriosis further improves the chances of becoming pregnant. After two failed IVF-ICSI cycles, it is preferable to suggest surgery because adding more cycles has minimal advantage on CPR, and the possibility of spontaneous pregnancy could reinforce this approach after surgery, even in individuals who have failed IVF-ICSI [76].

### 4.5. The Role of Surgery in the Infertility Management

Concerning our observations, our review underlines the lack of data, especially from randomized trials, to determine the surgical management of women with deep endometriosis and associated infertility. 

Starting from the data in the literature reporting that the spontaneous fertility rate might range from 2 to 10% in women with severe deep endometriosis, surgery seems to improve fertility. However, evident recommendations cannot be provided due to confounding factors; most studies in this review show a postoperative spontaneous pregnancy rate ranging from about 20% to more than 60%. The highest values came from two high-volume, single-surgeon centers with a low risk of major complications in general [58,69]. These findings may not be broadly applicable, but they suggest that deep endometriosis surgery should be limited to subspecialized centers where results are supervised to provide appropriate data for decision making. Although the available data are of poor quality, it appears that surgery for deep endometriosis improves the spontaneous pregnancy rate. 

Deep endometriosis grade III complications surgery occurred with a mean of approximately 7–8%. The studies included in our review do not only investigate fertility outcomes in women who have had complications and do not specify which women have had complications among those who have achieved pregnancy. Instead, even if it was impossible to perform any statistical analysis due to the heterogeneity of the data, a trend is evident that the centers with fewer (<5%) major complications (such as anastomotic leakage or permanent bladder atony) had a higher rate of spontaneous pregnancies. This finding is in line with the study of Ferrier, who suggests that patients with grade III complications be referred immediately to ART for higher chances of CPR [79]. 

The data on local excision vs. segmental resection do not enable us to reach a conclusion on the best technique in terms of pregnancy outcome; however, the technique that allows for appropriate deep endometriosis excision while having the lowest rate of complications will always be the clinical option. Although the only RCT on this topic shows no difference between the two groups regarding the pregnancy rate [69], our review reports a trend showing that the spontaneous pregnancy rate is higher in patients who undergo conservative techniques (46.2 vs. 36.7%).

Various studies on fertility in women with deep endometriosis discover a trend toward a possible benefit of combining surgery and MAR in increasing the chances of becoming pregnant [47,85]. The clinical evaluation of these patients should include the entire situation, including local IVF treatment options, success rates, public funding possibilities, and the inherent risk of surgical complications in the regional endometriosis center. In the absence of RCTs, comparative studies with control groups are required to assess the effects of deep endometriosis surgery before ART. There was only one such study included in our review [47]. Although the findings suggested that surgery could improve IVF treatment outcomes, more research is needed to determine the role of this approach. The available data suggest that surgery prior to IVF may be considered in cases of concomitant pain and repeated IVF failure after careful clinical evaluation and collaboration with the regional endometriosis center.

The management of women with severe deep endometriosis who want to become pregnant is currently under debate. The lack of a randomized trial comparing primary surgery to first-line IVF treatment allows for various conclusions from the currently available studies, including non-comparative retrospective cohorts. On the one hand, many studies back the use of first-line ART, while on the other hand, many authors defend first-line surgery [44].

The randomized trial EFFORT has been designed and is presently running with a focus on determining whether surgical treatment or first-line IVF should be considered in women with rectal deep endometriosis and pregnancy desire (NCT04610710). A total of 352 women between 18 and 38 will be randomly assigned in this study to receive surgical treatment (shaving, disc excision, or segmental resection) or ART treatment (at least two IVF or ICSI if not pregnant after the first cycle) [90]. 

Without question, therapeutic choices should be personalized. The patient’s age, preferences, and priorities; past treatments for the disease; ovarian reserve; tubal patency status; the presence of large or bilateral endometriomas; associated infertility factors (such as the quality of the partner’s semen); the duration of infertility; the severity of symptoms; the characteristics of various healthcare services; as well as reimbursement policies may all have an impact on the final decision. Women with deep endometriosis who have already undergone surgical procedures or ART interventions should receive information on the prospective advantages of treatments individually. The occurrence of pelvic pain in addition to infertility, particularly in women who are not interested in ART procedures, may tip the scale in favor of surgery, as this would not only reasonably enhance the likelihood of pregnancy but would also temporarily decrease the intensity of pain symptoms, allowing an appropriate quality of life during intervals of natural pregnancy seeking. 

## 5. Conclusions

The lack of information from RCTs to recommend the surgical treatment of women with deep endometriosis and related infertility is highlighted by our review. Although some studies have found a promising benefit of surgery on fertility outcomes, more studies are necessary to define fertility outcomes according to surgical procedures and whether surgical management should be used as the first intervention or only in cases where medically assisted reproduction treatment has failed. Positive benefits cannot be excluded, but the possibility of major complications needs to be considered, and the procedures should be performed in subspecialized centers.

## Figures and Tables

**Figure 1 jcm-11-06727-f001:**
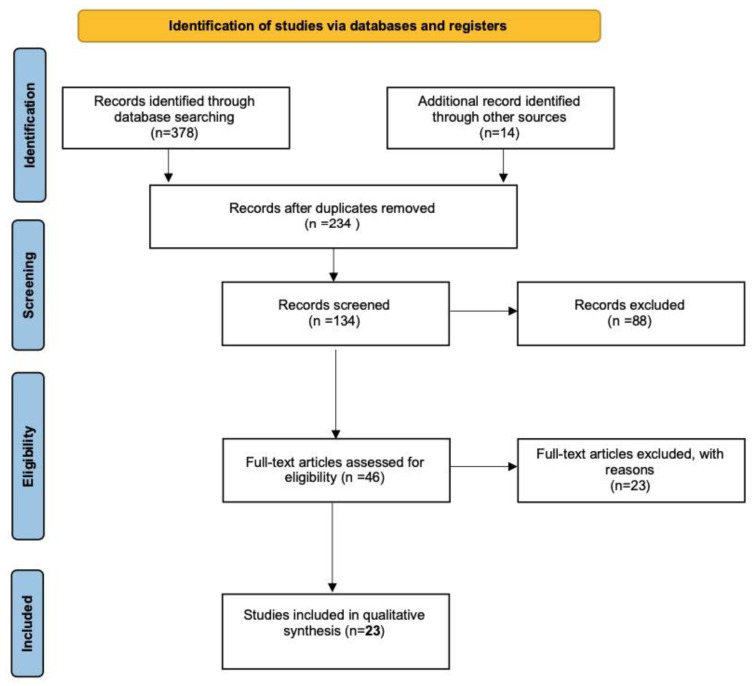
Flow diagram of study identification and selection.

**Table 1 jcm-11-06727-t001:** Main characteristics of considered studies and fertility outcomes after surgery for deep endometriosis: spontaneous and MAR.

Article	Study Design	Patients Studied for PR n	Endometriosis Localization (Main Site)	Documented Infertility before Surgery %	WTC %	TPR % (SP + MAR)	TPR in WTC %	Llive Birth %
Kovoor et al., 2010 [49]	Retrospective observational	10	Bladder	100%	100%	60%	60%	NR
Kavallaris et al., 2011 [50]	Retrospective observational	30	Rectovaginal and bowel	NR	56.6%	36.6%	64.7%	81.8%
Daraı et al., 2011 [51]	RCT	52	Colorectal	44.3%	53.8%	53.8%	39.3%	NR
Douay-Hauser et al., 2011 [52]	Retrospective controlled	75	No specific involvement	100%	100%	18.7%	18.7%	NR
Meuleman et al., 2011 [53]	Retrospective observational	28	Colorectal	NR	100%	46.4%	46.4%	NR
Rosznyai et al., 2011 [54]	Retrospective observational	9	Bladder and ureteral	NR	100%	66.6%	66.6%	NR
Jelenc et al., 2012 [55]	Retrospective observational	14	Colorectal	NR	100%	71.4%	71.4%	NR
Meuleman et al., 2014 [56]	Prospective controlled	148	Colorectal	NR	NR	50.7%	NR	81.3%
Tarjanne et al., 2014 [57]	Retrospective observational	164	Colorectal	NR	53.7%	25%	46.5%	100%
Malzoni et al., 2016 [58]	Retrospective observational	72	Colorectal	NR	100%	69.4%	69.4%	68%
Soriano et al., 2016 [59]	Retrospective observational	42	Bladder	64.3%	100%	80.9%	85.7%	100%
Saavalainen et al., 2016 [60]	Retrospective observational	28	Bladder	NR	NR	64.2%	NR	NR
Centini et al., 2016 [61]	Retrospective observational	115	No specific involvement	100%	NR	54.8%	NR	77.7%
Ballester et al., 2017 [62]	Prospective observational	60	Colorectal	90%	100%	60%	60%	58.3%
Millochau et al., 2017 [63]	Retrospective observational	9	Colorectal	NR	NR	66.6%	NR	83%
Bendifallah et al., 2017 [47]	Retrospective controlled	55	Colorectal	100%	100%	49.1%	49.1%	100%
Stochino Loi et al., 2017 [64]	Retrospective controlled	180	No specific involvement	75%	75%	74.4%	NR	72.4%
Timoh et al., 2018 [65]	Retrospective observational	34	Bladder with and without posterior compartment involvement	50%	73.5%	50%	68%	76.5%
Hudelist et al., 2018 [66]	Prospective observational	61	Bowel	54.5%	100%	63.9%	63.9%	76.9%
Rubod et al., 2019 [48]	Retrospective observational	152	No specific involvement	100%	100%	49.3%	49.3%	NR
Arfi et al., 2019 [67]	Retrospective observational	118	No specific involvement	55.9%	100%	39%	39%	78.2%
Zhang et al., 2022 [68]	Retrospective observational	55	No specific involvement	100%	100%	61.8%	61.8%	82.3%
Roman et al., 2022 [69]	RCT	37	Colorectal	NR	100%	83.8%	83.8%	NR
Total		1548		79.7%	86.6%	52.6% 95% CI (49.7–63)	53% 95% CI (40.2–77.4)	78.3% 95% CI (60.8–80.2)

Legend: PR pregnancy rate; WTC wish to conceive; TPR total pregnancy rate; SP spontaneous pregnancies; MAR medically assisted reproduction; NR not reported.

**Table 2 jcm-11-06727-t002:** Pregnancy rates in infertile women WTC.

Article	Infertile Women WTC n	TPR %	SP %	MAR %
Kovoor et al., 2010 [49]	10	60%	50%	10%
Daraı et al., 2011 [51]	15	33.3%	26.7%	6.6%
Douay-Hauser et al., 2011 [52]	75	18.7%	18.7%	0%
Meuleman et al., 2011 [53]	28	46.4%	28.6%	17.8%
Jelenc et al., 2012 [55]	14	71.4%	57.1%	14.3%
Malzoni et al., 2016 [58]	72	69.4%	61.1%	8.3%
Ballester et al., 2017 [62]	54	66.6%	0%	66.6%
Soriano et al., 2016 [59]	27	70.4%	18.5%	51.9%
Bendifallah et al., 2017 [47]	55	49.1%	0%	49.1%
Timoh et al., 2018 [65]	17	52.9%	35.3%	17.6%
Rubod et al., 2019 [48]	152	49.3%	0%	49.3%
Hudelist et al., 2018 [66]	61	63.9%	42.6%	21.3%
Zhang et al., 2022 [68]	55	61.8%	43.6%	18.2%
Total	635	53% 95% CI (46.3–63.4)	22.7% 95% CI (15.3–57.2)	30.3% 95% CI (7.6–58)

Legend: WTC wish to conceive; TPR total pregnancy rate; SP spontaneous; MAR medically assisted reproduction.

**Table 3 jcm-11-06727-t003:** Studies on PR, postoperative complications, and recurrences after colorectal surgery for deep endometriosis.

Article Focusing on Bowel Involvement	Number of Patients n	Women Studied for PR n	TPR %	SP %	MAR %	Type of Treatment	Clavien–Dindo Complications Grade % and Recurrences %
Kavallaris et al., 2011 [50]	55	30	36.6%	23.3%	13.3%	Combined LPS/vaginal technique en bloc excision of the posterior vaginal wall, rectovaginal septum, and a part of the rectosigmoid	Grade II: 1.8% Grade III: 5.4% 25.5% bladder atony requiring self- catheterization for a mean of 3 months 3.6% anastomotic leakage 3.6% bowel recurrence reoperation Recurrences NR
Darai et al., 2011 [51]	52	52	53.8%	11.5%	42.3%	LPS or LPT colorectal resection LPS: 17.3% LPT 28.8%	Grade I: 3.8% Grade II: 42.3% Grade III: 23% Recurrences NR
Meuleman et al., 2011 [53]	45	28	46.4%	28.6%	17.8%	CO2 laser lps radical deep endometriosis excision followed by bowel resection and reanastomosis	Grade III: 2.2% 2.2% bladder atony required self- catheterization for 10 weeks 2.2% LPS salpingectomy for hydrosalpinx 1 year after the surgery Recurrences 4.4%
Jelenc et al., 2012 [55]	56	14	71.4%	57.1%	14.3%	LPS segmental bowel resection and reanastomosis	Grade III: 10.7% 5.4% anastomotic leakage 3.6% rectovaginal fistula 1.8% LPT for bleeding in pelvic region 3.6% anastomotic stricture Recurrences NR
Meuleman et al., 2014 [56]	203	148	50.7%	20.9%	29.7%	Radical excision of moderate–severe endometriosis with or without LPS bowel resection and reanastomosis	Grade I–II: 4.4% Grade III: 2% 1.3% anastomotic leakage 1.3% rectovaginal fistula 1% bladder atony 0.98% bladder leakage Recurrerences: 3.9%
Tarjanne et al., 2014 [57]	164	164	25%	NR	NR	LPS or LPT colorectal resection	Grade III: 8.5% 2.4% anastomotic leakage 1.8% ureteral fistulae 1.2% rectovaginal fistula 0.6% bladder atony for 3 months Recurrences: 6.7%
Malzoni et al., 2016 [58]	248	72	69.4%	61.1%	8.3%	LPS bowel segmental resection and deep endometriosis excision	Grade II: 2.8% Grade III: 4.8% Grade IV: 0.4% 1.6% anastomotic leakage 2.4% rectovaginal fistula 0.8% severe peritonitis Recurrences NR
Ballester et al., 2017 [62]	60	60	60%	0%	60%	LPS surgery: rectal shaving, full thickness disc excision or segmental colorectal resection	NR
Millochau et al., 2017 [63]	21	9	66.6%	22.2%	44.4%	LPS rectal disc excision or short segmental resection	Grade II: 19% Grade III: 28.6% 19% rectovaginal fistula Recurrences NR
Bendifallah et al., 2017 [47]	55	55	49.1%	0%	49.1%	LPS surgery: rectal shaving, full thickness disc excision or segmental colorectal resection	NR
Hudelist et al., 2018 [66]	134	61	63.9%	42.6%	21.3%	LPS disc and limited nerve- and vessel-sparing segmental resection	Grade I: 7.5% Grade II: 2.2% Grade III: 6% 6.7% temporary bladder atony 1.5% anastomotic leakage 0.7% rectovaginal fistula Recurrences NR
Roman et al., 2022 [69]	55	37	83.8%	NR	NR	LPS surgery: rectal shaving, full thickness disc excision or segmental colorectal resection	Grade III: 20% 3.6% rectovaginal fistula 1.8% bladder fistula 12.7% bladder atony required self- catheterization for 30 days Recurrences 3.6%
Total	1148	730	50.3% 95% CI (47.2–65.6)	24.9% 95% CI (8.1–34.5)	30.8% 95% CI (19–41)		Grade I: 1.2% Grade II: 4.45% Grade III: 7.45% Grade IV: 0.09% 2% anastomotic leakage 5.2% bladder atony required self- catheterization 2.4% rectovaginal fistula Recurrences 4.8%

Legend: PR pregnancy rate; TPR total pregnancy rate; SP spontaneous pregnancies; MAR medical assisted reproduction; OMA ovarian endometrioma; NR not reported; LPS laparoscopy; LPT laparotomy.

**Table 4 jcm-11-06727-t004:** PR in women who underwent conservative surgery (shaving and/or discoid resection) or segmental resection.

Studies	Patients n	DR and/or Shaving %	TPR in Conservative Surgery %	SP in Conservative Surgery %	MAR in Conservative Surgery %	SR %	TPR in SR %	SPR in SR %	MAR in SR %	TPR in Patients WTC %
Kavallaris et al., 2011 [50]	30	0	0	0	0	100%	36.6%	23.3%	13.3 %	64.7%
Daraı et al., 2011 [51]	52	0	0	0	0	100%	53.8%	11.5%	42.3%	39.3%
Meuleman et al., 2011 [53]	28	0	0	0	0	100%	46.4%	28.6%	17.8%	46.4%
Jelenc et al., 2012 [55]	14	0	0	0	0	100%	71.4%	57.1%	14.3%	71.4%
Tarjanne et al., 2014 [57]	164	0	0	0	0	100%	25%	NR	NR	46.5%
Malzoni et al., 2016 [58]	72	0	0	0	0	100%	69.4%	61.1%	8.3%	69.4%
Ballester et al., 2017 [62]	9	22.2%	100%	0%	100%	77.7%	42.8%	0%	42.8%	55.5%
Hudelist et al., 2018 [66]	61	18%	63.6%	54.5%	9%	82%	64%	40%	24%	63.9%
Total	430	3%	69.2%	46.2%	23%	97%	45%	36.7%	21.4%	56.8%

Legend: PR pregnancy rate; DR discoid resection; TPR total pregnancy rate; SP spontaneous pregnancies; MAR medical assisted reproduction; SR segmental resection; WTC wish to conceive; NR not reported.

**Table 5 jcm-11-06727-t005:** Studies on PR, postoperative complications, and recurrences after surgery for bladder deep endometriosis.

Article Focusing on Bladder Involvement	Number of Patients n	Women Studied for PR n	TPR %	SP %	MAR %	Type of Treatment	Clavien–Dindo Complications Grade % and Recurrences %
Kovoor et al., 2010 [49]	21	10	60%	50%	10%	LPS partial cystectomy or partial thickness excision of the detrusor muscle	Grade III: 19% 9.5% vescicovaginal fistula Recurrences NR
Rozsnyai et al., 2011 [54]	30	9	66.6%	55.5%	11.1%	LPS partial cystectomy or partial thickness excision of the detrusor muscle and/or ureteral surgery	Grade II: 6.6% Grade III: 13.3% 6.6% bladder atony required self- catheterization for >6 months 3.3% vescico vaginal fistula Recurrences NR
Soriano et al., 2016 [59]	69	42	80.9%	38.1.%	42.8%	LPS partial cystectomy or LPS partial thickness excision of the detrusor muscle	Grade III: 2.9% Recurrences 2.9%
Saavalainen et al., 2016 [60]	53	28	64.2%	25%	39.3%	LPS and/or partial cystectomy or LPS partial thickness excision of the detrusor muscle	Grade I: 3.8% Grade II: 39.6% Grade III: 9.4% Recurrences NR
Timoh et al., 2018 [65]	34	34	50%	35.3%	14.7%	LPS bladder resection associated with or without posterior DE resection	Grade I: 8.8% Grade II: 5.8% 5.8% bladder atony required self-catheterization Recurrences NR
Total	207	123	65.9% 95% CI (54.5–74.2)	36.6% 95% CI (30.2–51)	29.3% 95% CI (9.5–37.7)		Grade I: 2.4% Grade II: 12% Grade III: 7.2%

Legend: PR pregnancy rate; TPR total pregnancy rate; SP spontaneous pregnancies; MAR medical assisted reproduction; OMA ovarian endometrioma; NR not reported; LPS laparoscopy; LPT laparotomy.

**Table 6 jcm-11-06727-t006:** Studies on PR postoperative complications and recurrences after surgery for deep endometriosis without a specific involvement.

Article Without Specific Involvement	Number of Patients n	Women Studied for PR n	TPR %	SP %	MAR %	Type of Treatment	Clavien–Dindo Complications Grade % and Recurrences %
Douay-hauser et al., 2011 [52]	75	75	18.7%	18.7%	0%	LPS/LPT excision or coagulation of all visible endometriosis lesions. Segmental LPT resection in case of bowel involvement	Grade III: 8% 2.6% ureteral injuries 1.38% bowel perforation 1.38% abdominal wall abscess 1.38% ovarian abscess 1.38% anastomotic fistula Recurrences NR
Centini et al., 2016 [61]	115	115	54.8%	26.1%	28.7%	Laparoscopic treatment of anterior, posterior, and lateral compartment	Grade III: 7.8% 1.7% vescicovaginal fistula 0.9% urinoma 0.9% rectovaginal fistula Recurrences 16.5%
Stochino loi et al., 2017 [64]	180	180	74.4%	41.1%	33.3%	Laparoscopic treatment of anterior, posterior, and lateral compartment.	NR
Rubod et al., 2019 [48]	152	152	49.3%	0%	49.3%	Excision of deep posterior endometriosis	NR
Arfi et al., 2019 [67]	118	118	39%	20.4%	18.6%	LPS excision without bowel involvement	NR
Zhang et al., 2022 [68]	55	55	61.8%	43.6%	18.2%	LPS excision with and without bowel involvement (shaving or segmental resection)	NR
Total	695	695	52.7% IC95% (34.2–65.1)	23.9% IC95% (12.1–37)	28.8% IC95% (11.4–38)		Grade III: 7.9% 1% rectovaginal fistula

Legend: PR pregnancy rate; TPR total pregnancy rate; SP spontaneous pregnancies; MAR medical assisted reproduction; OMA ovarian endometrioma; NR not reported; LPS laparoscopy; LPT laparotomy.

## Data Availability

All data generated or analyzed during this study are included in this published article.

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
