# Peer review of "Deep Endometriosis and Infertility: What Is the Impact of Surgery?"

_jcm, 2022, doi:10.3390/jcm11226727_

Round 1

Reviewer 1 Report

This title looks very attractive and valuable question for both patients and healthcare professionals. However, I cannot find the answer of that question in this article. It is true that performing high quality studies for evaluating the impact of surgery have a lot of hurdles. Therefore, the experts who made ESHERE guideline suggest only by GPP (good practice point). If the authors want to answer this question, they must organize the articles much more readable and meaningful. However, this article fails to find brand new findings and brand new style of review article. If the authors want to make this review article more readable and valuable, they should reorganize discussion part. It will be better to make subtitles according to the contents in discussion section. 

Author Response

Point 1: This title looks very attractive and valuable question for both patients and healthcare professionals. However, I cannot find the answer of that question in this article. It is true that performing high quality studies for evaluating the impact of surgery have a lot of hurdles. Therefore, the experts who made ESHERE guideline suggest only by GPP (good practice point). If the authors want to answer this question, they must organize the articles much more readable and meaningful. However, this article fails to find brand new findings and brand new style of review article. If the authors want to make this review article more readable and valuable, they should reorganize discussion part. It will be better to make subtitles according to the contents in discussion section.

Response 1: Thanks for your helpful suggestions.

Our review planned to systematically collect all the articles defining the impact of surgery on the patient with DIE. Unfortunately, to definitively answer the title question, we have to wait for the results of the ENDOFERT and EFFORT RCTs. Hoping to make the manuscript more readable and meaningful, following your advice, we divided the discussion by emphasizing the role of surgery based on the location of the DIE. We also highlighted the role of surgery before and/or after ART.

Reviewer 2 Report

In DIE patients wishing to get pregnant, deciding whether to prioritize surgery or ART has been challenging because of the lack of randomized trials. The manuscript is well written and well discussed by reviewing plenty of previous articles and sheds light on clinicians' choice of treatment options for DIE patients who desire pregnancy after infertility or infertility treatment periods. 

The manuscript allows us to understand the articles that have been reported so far. The reviewer thinks no changes are required.

Specific comments

#1. Please check the number of articles in Figure 1. The reviewer is afraid that the Screening step includes the wrong number(s).

Author Response

Point 1: In DIE patients wishing to get pregnant, deciding whether to prioritize surgery or ART has been challenging because of the lack of randomized trials. The manuscript is well written and well discussed by reviewing plenty of previous articles and sheds light on clinicians' choice of treatment options for DIE patients who desire pregnancy after infertility or infertility treatment periods.

The manuscript allows us to understand the articles that have been reported so far. The reviewer thinks no changes are required.

Specific comments

#1. Please check the number of articles in Figure 1. The reviewer is afraid that the Screening step includes the wrong number(s).

Response 1: Thanks for your positive comment on our review. We apologize for the mistake in figure n1. We have corrected the wrong steps.
